# The Impact of Meaningful Use and Electronic Health Records on Hospital Patient Safety

**DOI:** 10.3390/ijerph191912525

**Published:** 2022-09-30

**Authors:** Kate E. Trout, Li-Wu Chen, Fernando A. Wilson, Hyo Jung Tak, David Palm

**Affiliations:** 1Department of Health Sciences, School of Health Professions, University of Missouri, 329 Clark Hall, Columbia, MO 65211, USA; 2Matheson Center for Health Care Studies, University of Utah, Salt Lake City, UT 84108, USA; 3Department of Health Services Research and Administration, College of Public Health, University of Nebraska Medical Center, Omaha, NE 68198, USA

**Keywords:** patient safety indicators, reimbursement incentive, patient safety, health services, electronic health records

## Abstract

The HITECH Act aimed to leverage Electronic Health Records (EHRs) to improve efficiency, quality, and patient safety. Patient safety and EHR use have been understudied, making it difficult to determine if EHRs improve patient safety. The objective of this study was to determine the impact of EHRs and attesting to Meaningful Use (MU) on Patient Safety Indicators (PSIs). A multivariate regression analysis was performed using a generalized linear model method to examine the impact of EHR use on PSIs. Fully implemented EHRs not attesting to MU had a positive impact on three PSIs, and hospitals that attested to MU had a positive impact on two. Attesting to MU or having a fully implemented EHR were not drivers of PSI-90 composite score, suggesting that hospitals may not see significant differences in patient safety with the use of EHR systems as hospitals move towards pay-for-performance models. Policy and practice may want to focus on defining metrics and PSIs that are highly preventable to avoid penalizing hospitals through reimbursement, and work toward adopting advanced analytics to better leverage EHR data. These findings will assist hospital leaders to find strategies to better leverage EHRs, rather than relying on achieving benchmarks of MU objectives.

## 1. Introduction

The United States has made a significant investment in the adoption and use of Health Information Technologies (HIT) by providing over 35 billion dollars of support through the Health Information Technology for Economic and Clinical Health (HITECH) Act [1]. Into the 21st century, the HITECH Act aimed to leverage digitized health records, known as Electronic Health Records (EHRs), to improve the efficiency, quality, patient safety, and health outcomes by standardizing health data, establishing better care coordination across providers, and ultimately providing clinical decision support [1]. Since 2011, the Medicare and Medicaid EHR Incentive Programs (now known as the Promoting Interoperability Programs) has provided financial incentives to providers and healthcare organizations that demonstrate “Meaningful Use” (MU) with certified EHR systems. MU is achieved by meeting a set of objectives specified by Centers for Medicare and Medicaid Services (CMS) and the Office of the National Coordinator for Health IT (ONC) that aim to improve patient safety and quality outcomes. Some studies show EHRs have achieved gains in patient satisfaction, clinical outcomes, risk management, and decision support [2,3,4,5,6,7]. However, the literature has provided mixed empirical evidence regarding the ability of EHRs to achieve improvements in other outcomes, including patient safety, quality, and cost-efficiency [8,9,10,11].

Patient safety accounted for 25 percent of the total performance score (TPS) in the fiscal year 2020 of the Hospital Value-Based Purchasing (HVBP) Program. The Agency for Healthcare Research and Quality’s (AHRQ) Patient Safety Indicators (PSIs) were developed as metrics for likely preventable patient complications and adverse events following surgeries and procedures, which could provide an opportunity for improvement in healthcare delivery. Individual PSIs and PSI-90 composite scores have been used as metrics in the TPS of the value-based purchasing model, and an updated PSI-90 (6th version) composite score will be incorporated into HVBP in fiscal year 2023 [12]. Evaluating the 5th version of the PSI-90 composite and individual PSIs can inform the effectiveness of the MU of EHRs in hospital settings; although, patient safety and MU attestation have been understudied, making it difficult to determine if the specific set of MU objectives chosen by CMS are drivers of individual PSIs and the PSI-90 composite scores.

Quantitative research that evaluates the impact of EHRs and MU attestations on patient safety outcomes is scarce; although, the evidence suggests that EHRs and the MU may not be significant drivers of patient safety thus far. The majority of studies demonstrating a positive impact of EHRs on patient safety only focus on specific functionalities with the use of EHRs, such as clinical decision support or computerized provider order entry, but not actually attesting to MU [8]. One study found that EHRs and HITs had little to no association with hospital readmission rates [13]. To our knowledge, only three studies investigate the impact of MU on patient safety outcomes [11,14,15]. Another study found that Stage 1 MU capable EHR systems were associated with improvements on 3 of 8 patient safety measures with 7% to 11% lower rates of adverse events [15]. However, this study has a major limitation where Stage 1 MU was determined by classifying functionalities that could potentially meet Stage 1 MU, but not actually attesting to MU. EHR systems were classified as meeting 2011 MU functionalities using information collected in 2007, but evaluated patient safety estimates from 2008 to 2010. This kind of gap could misclassify many hospitals that adopted the Stage 1 functionalities for MU after 2007, which is likely because of the steep increase in EHR adoption that took place in 2009 and after with the implementation of the ACA and the HITECH Act [16].

To the best of our knowledge, no study has determined the impact of hospital MU attestation on PSI-90 composite scores, which will be informative as pay-for performance models are revising the PSI-90 composite score and its role in HVBP. The objective of this study is to determine the impact of hospital MU attestation and EHR use on patient safety outcomes, including the PSI-90 composite score.

## 2. Methods

### 2.1. Data

The inpatient hospitalization data used in this study were from Healthcare Cost and Utilization Project 2013 State Inpatient Databases of Florida, Nebraska, New York, and Washington to examine the impact of EHRs on patient safety outcomes. One state was randomly chosen from each of the census regions to be geographically representative of a national sample. The 2013 American Hospital Association (AHA) annual survey data and CMS MU attestation records were used to gather hospital characteristics and information regarding their EHR systems. The final sample included 349 hospitals that provided information about their EHR systems. 

### 2.2. Calculating PSIs

Patient safety was measured using AHRQ’s PSIs, which are a set of indicators providing information on patient’s potential hospital complications and adverse events following surgeries and procedures. The AHRQ PSI software version 5.0 and SAS version 9.4 statistical software were used to determine the hospital-level risk-adjusted standardized rates for 8 patient safety indicators (PSIs), and the PSI-90 composite score. The AHRQ risk-adjustment software uses a complex algorithm to adjust for patient characteristics, severity of illness, and 25 comorbidities as covariates [17]. It was used in conjunction with the 2013 population file to produce risk-adjusted rates based on the general population at risk during the year 2013. In addition, the PSI-90 composite scores were calculated using the AHRQ PSI software to determine the overall impact on patient safety by calculating the weighted average of the reliability-adjusted observed-to-expected ratios. The component measures are expressed as a ratio to the reference population rate, where a provider would have a composite rate of 1 if the risk adjusted ratio component score is the same as the reference population. Composite scores of 1 represent the same quality as the national average.

### 2.3. Analytical Approach

MU attestation was determined from the CMS attestation records which identifies the stage of MU attested to by the hospital, the incentives they received, and the years they attested. The 2013 AHA annual survey data was used to identify information regarding the use of an EHR system. Hospitals were categorized into three groups for this study: (1) attesting to MU with their EHRs, (2) having a fully implemented EHR but have not attested to MU, and (3) having partially implemented EHR or no EHR system (see Figure 1).

Data summary statistics and bivariate analyses were performed to examine the difference in outcome variables and explanatory variables, including analysis of variance (ANOVA) and chi squared tests. The multivariate regression analyses utilized the generalized linear model (GLM) method with log link function and gamma family distribution to examine the impact of EHR use on the individual PSIs and the PSI-90 composite scores. Safety-related adverse events, if measured using with a Poisson parameter (ex. mean rate for patients) across each facility, should be considered gamma distributed [15,18,19]. This is consistent with the previous literature where PSI measures are rate variables, and each PSI was modeled as a nonlinear regression model with a log link function and gamma distribution using a GLM model [15]. The model coefficient represents the semi-elasticity, where the dependent variable changes by 100 * (coefficient) percent for a one-unit increase in the independent variable while all other variables in the model are held constant. The final model adjusted for minor teaching hospital status, major teaching hospital status, for profit status, nurse-to-staffed bed ratio, state, and staffed beds. All analyses were conducted using Stata/IC v.14.1.

## 3. Results

### 3.1. Hospital Characteristics

The majority of the hospitals attested to MU Stage 1 with their EHR systems (82.2%), followed by having partially implemented or no EHR system (9.2%) and having a fully implemented EHR system but does not attest to MU (8.6%). The majority of hospitals had 100–299 beds, had non-profit status, were not teaching hospitals, located in a metropolitan area, from New York, and had an average nurse to bed ratio of 1.73. There were significant differences in the number of staffed hospital beds and state across EHR use groups (Table 1).

### 3.2. Impact of EHR Use on Patient Safety

Among EHR groups, there were significant differences in 7 patient safety outcomes (Table 2). Partially implemented or no EHRs had a higher mean incidence for the following patient safety outcomes: low-mortality diagnosis related groups (DRGs) with a mean death rate of 1.04 deaths per 1000 patients; postoperative physiologic and metabolic derangement rate with a mean incidence of 2.20 incidence per 1000 patients; serious blood clots after surgery with a mean incidence of 9.21 incidence per 1000 patients; and wounds split open after surgery with a mean incidence rate of 5.90 incidence per 1000 patients. Fully implemented EHRs that did not attest to MU had the highest mean incidence rate of postoperative sepsis with a mean of 19.34 incidence per 1000 patients, and breathing failure after surgery with a mean incidence of 9.25 incidence per 1000 patients.

Table 3 shows the impact of EHRs on patient safety outcomes after adjusting for important confounders. Fully implemented EHRs that did not attest to MU had a significant decrease in adverse events on 3 patient safety outcomes, and EHRs that attested to MU had a significant decrease in adverse events on 2 patient safety outcomes. EHRs that attested to MU had a significant decrease in adverse events on postoperative physiologic and metabolic derangement rate and in perioperative pulmonary embolism or deep vein thrombosis rate. However, there was no significant impact of attesting to MU or having a fully implemented EHR not attesting to MU on the PSI-90 composite score compared partially implemented or no EHR systems.

The effect of fully implemented EHRs that did not attest to MU were larger than those EHRs that attested to MU. The death rate in low-mortality DRGs decreased by 291% for those hospitals with a fully implemented EHR system that did not attest to MU compared to hospitals with a partially implemented or no EHR, indicating a positive impact. The effect was diminished among hospitals attesting to MU, decreasing by 93% compared to hospitals with a partially implemented or no EHR, although not statistically significant. This same effect between groups was observed among postoperative physiologic and metabolic derangement rate (242% verses 119%). Postoperative wound dehiscence rate decreased by 193% for those hospitals with a fully implemented EHR system that did not attest to MU, respectively. Among hospitals attesting to MU perioperative pulmonary embolism or deep vein thrombosis rate decreased by 89%. Although, there was not a significant impact observed among fully implemented EHRs that did not attest to MU compared to partially implemented or no EHR system.

## 4. Discussion

The impact of MU attestation on patient safety has been understudied, making it difficult to determine if the specific set of MU objectives have had a positive impact on outcomes. It will be important to study the impact of MU attestation on the ability to achieve these intended outcomes envisioned by the Affordable Care Act (ACA) [20] and HITECH Act to direct policies and implementation related to the next phases of Promoting Interoperability Programs. Our results are consistent with the literature and suggests that critical evaluation is needed for EHR implementation related to specific patient safety metrics [11]. The most recently published study in 2020 evaluated the 7 separate MU performance measures to determine the association with achieving gains in patient safety. Even among hospitals that had EHR implementation above MU performance thresholds, the quantile regression results showed there was not a consistent association with gains in patient safety measures used in the HVBP Program [11]. We found that hospitals attesting to MU with their EHR systems improved only 2 patient safety outcomes. EHR use did not have a significant impact on PSI composite scores in 2013 [15]. This evidence suggests that hospitals that invest in adopting EHR systems may not see significant improvements in their PSI-90 composite scores, especially as CMS moves toward pay-for-performance models that incorporate the PSI-90 in total performance scores (TPSs). The hospitals with low TPSs will need to focus on other factors and strategies that may significantly impact the PSI-90 composite score to avoid reductions in reimbursement, such as process improvement and staff training. ONC will need to focus on functionalities and advanced analytics that result in improvements in safety and quality outcomes. More research is needed to determine strategies that significantly improve the PSI-90 composite score for providers. Furthermore, policy makers may want to focus on specific patient safety indicators that are highly preventable in payment models to avoid penalizing hospitals through reimbursement, rather than incorporating the PSI-90 composite score.

Surgery is one of the leading causes of blood clot problems, resulting in conditions such as pulmonary embolism or deep vein thrombosis [21]. EHRs have the potential to improve patient safety, particularly for surgical care by providing timely and meaningful health information that could prevent medical errors or reduce the impact of errors that have been made. Our results show that hospitals that had MU EHR systems had significantly decreased risk of perioperative pulmonary embolism or deep vein thrombosis by 89% relative to those hospitals with a partially implemented or no EHR system. Venous thromboembolism and pulmonary embolism are often lethal with 25% of cases resulting in sudden death, and 1-week survival rate is 71% after a pulmonary embolism [22]. Survivors of both conditions may experience serious and costly long-term complications [23]. The appropriate medication can be given before and after major surgeries to greatly reduce the risk and prevent blood clots with low, fixed doses of anticoagulant drugs [24]. We did not find a significant difference for hospitals with fully implemented EHRs that did not attest to MU, suggesting the objectives of Stage 1 effectively leveraged EHRs to better prevent perioperative pulmonary embolism and deep vein thrombosis. More research needs to be conducted to determine the effectiveness of specific functionalities and objectives in Stage 1 MU that contribute to the increased patient safety related to the prevention of perioperative pulmonary embolism or deep vein thrombosis, which could potentially be related to medication monitoring and decision support.

Furthermore, we found that both advanced EHR groups (fully implemented EHRs that did not attest to MU and hospitals that attested to MU) had a significant positive impact on reducing postoperative physiologic and metabolic derangement rate compared to hospitals that had a partially implemented or no EHR system. MU had a positive impact on these PSIs, but greater gains were observed among EHRs not attesting to MU. EHRs that did not attest to MU had significant positive impacts on reducing death rate in low-mortality DRGs and postoperative wound dehiscence rate. Our findings are consistent with previous literature where EHRs saw reductions in postoperative wound dehiscence [15]. These results may suggest that hospitals purchasing EHR systems without the MU incentives may face more pressure to receive the financial benefits, and may focus their efforts on leveraging their EHRs to improve selected outcomes to meet the needs of their practices. The postoperative physiologic and metabolic derangement rate decreased by 242% for those hospitals with a fully implemented EHR system that did not attest to MU compared to hospitals with a partially implemented or no EHR, and decreased by 199% among hospitals attesting to MU, respectively. Although not statistically significant for EHRs attesting to MU, this same effect between groups was observed among death rates in low-mortality DRGs (291% vs. 93% reductions), and postoperative wound dehiscence (193% vs. 86% reductions). More research is needed to determine the functionalities and drivers behind these gains in patient safety among hospitals not receiving MU incentives. Preliminary qualitative research suggests a “ceiling effect” may occur where physicians adopt EHR systems that meet the specified criteria without being optimal [25]. After appropriate training and becoming efficient in basic functionalities, physicians were likely to ignore more advanced EMR system functionalities [25]. It is possible that hospitals that do not receive incentives are better leveraging functionalities outside of the MU objectives to achieve these heightened gains in patient safety, which may need to be considered in adding to the MU objectives.

We did not find significant differences among most individual indicators between EHR groups. When using individual indicators, it is difficult to find significant variation among events that are rare, such as adverse patient safety events. Additionally, some single indicators face criticism for low predictability and reliability to determine hospital’s patient safety. For example, we did not find significant differences in respiratory failure between EHR groups. Diagnosis of respiratory failure often overlaps with airway management and are most often not preventable cases; thus, there are issues related to accuracy, reliability of physician diagnosis, and questionable preventability [26,27,28,29]. There is relatively little surgeons can do to minimize the risk of respiratory failure [28], but the most prominent non-modifiable risk factors for postoperative respiratory failure are advanced age, a major operation involving the torso, and substantial neurologic, cardiovascular, or pulmonary comorbidity [27,28,29]. Policy makers should take caution when using postoperative respiratory failure rate as a PSI to influence policy decisions, given concerns related to reliability, accuracy, and preventability. It is difficult to make inferences on overall patient safety using single indicators where meaningful composite scores may be more useful in determining the overall impact on patient safety [30]. It is also important to note that providers are reliant on the quality and timeliness of information provided in EHR systems, and the presence of advanced alerts that may indicate patients at risk of adverse events and supporting documentation for care protocols. Research needs to address these nuances in EHR use between healthcare organizations and its impact on individual PSIs.

### Limitations

Due to the cross-sectional nature to our study, we were not able to establish a causal relationship between EHR use and patient safety. Second, more studies need to be conducted with larger samples in the references group. Our study included 32 hospitals in the reference group, with hospitals containing partially implemented and no EHR systems. In our sample, 90.8% of hospitals had an EHR system, either that attested to MU or had a fully implemented EHRs. This is consistent with the national sample where about 94% of hospitals reported having a certified EHR in 2013 [31]. Our study provides a reference group with limited to no EHR functionality compared with two advanced EHR systems, one being the government standard for EHR use supported by the MU program. Not separating the other fully implemented EHRs out of the reference group may dilute the results and underestimate the observed impact of EHR use on outcomes. To our knowledge, this categorization has never been compared in the literature, but provides the opportunity to study outcomes among hospitals that attest to MU with the use of their EHRs (the government standard for EHR functionality) and those that have fully implemented EHR systems that do not attest to MU. Our study can act as a reference for future studies. National studies are needed to produce larger sample sizes, where these may be classified into two separate groups on varied EHR functionalities. However, the study provides valuable insight of the impact of MU and EHRs on patient safety outcomes.

Furthermore, we did not study the impact of specific EHR functionalities on patient safety outcomes, but rather a set of functionalities chosen for Stage 1 MU in 2013. Based on the results of this study, policymakers and hospital leaders may need to revisit the MU objectives for more stringent standards or additional functionalities may need to be adopted to improve patient safety, such as focusing on advanced analytics to better manage clinical operations and identify populations at risk. Future studies should account for other drivers of quality that could not be adjusted for in this study, including best practices of healthcare organizations, the presence of review committees, or participation in other programs linked to quality and patient safety. This study provides preliminary evidence on the impact of the government benchmark for EHR use with the MU program has had on achieving in patient safety in 2013.

## 5. Conclusions

This study has generated valuable insights into practical implications for hospital leaders seeking gains in patient safety with EHR systems and the Promoting Interoperability Programs. Our study demonstrates that hospitals attesting to MU with EHR systems improved 2 patient safety outcomes, but the MU benchmarks may not be stringent enough to produce consistent gains in patient safety. Hospital leaders should focus on better leveraging EHR functionalities outside of the MU objectives to improve health outcomes. Policy and practice may want to focus on adopting patient safety metrics that are highly preventable when incorporating patient safety into payment models to avoid penalizing hospitals through reimbursement, rather than incorporating the PSI-90 composite score.

## Figures and Tables

**Figure 1 ijerph-19-12525-f001:**
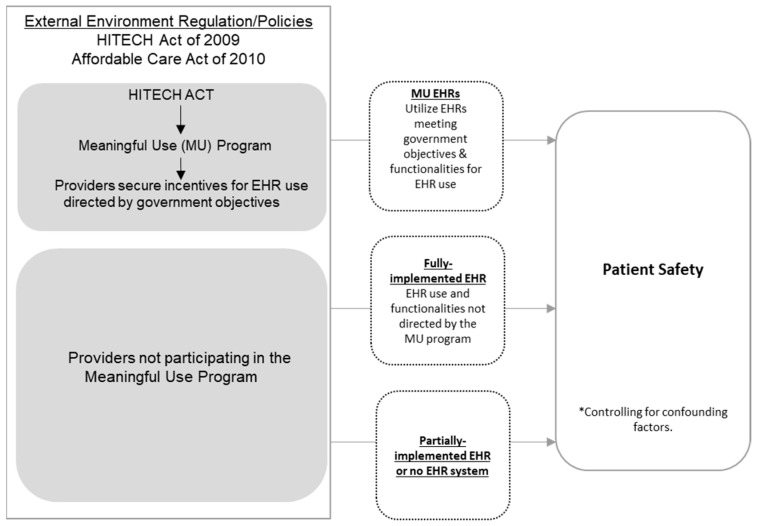
Study comparison groups regarding variation in EHR use. ***** Model controls for minor teaching hospital status, major teaching hospital status, for profit status, state, nurse to staffed bed ratio, and staffed beds.

**Table 1 ijerph-19-12525-t001:** Description of EHR use and Hospital Characteristics.

	Total Sample *n* (%) (*n* = 349)	Partially Implemented or No HER *n* (%) (*n* = 32)	Full-EHR without MU *n* (%) (*n* = 30)	EHR that Attests to MU *n* (%) (*n* = 287)	*p*-Value
**Hospital Characteristics**					
Number of staffed beds *Mean (SD)* <100 100–299 300–399 400–499 500 and greater	292.2 (17.2) 97 (27.8) 135 (38.7) 36 (10.3) 20 (5.7) 60 (17.5)	237.8 (73.5) 15 (46.9) 11 (34.4) 2 (6.25) 0 4 (12.5)	217.8 (79.2) 17 (56.7) 7 (23.3) 3 (10.0) 1 (3.3) 2 (6.7)	306.1 (17.4) 65 (22.7) 117 (40.8) 31 (10.8) 19 (6.6) 55 (19.2)	0.218 0.002
N (%) for profit	30 (8.6)	4 (12.5)	2 (6.7)	24 (8.4)	0.676
Teaching status Non-teaching Minor teaching Major teaching	199 (57.0) 110 (31.5) 40 (11.5)	21 (65.6) 8 (25.0) 3 (9.4)	19 (63.3) 7 (23.3) 4 (13.3)	159 (55.4) 95 (33.1) 33 (11.5)	0.687
**Location**					
State Florida Nebraska New York Washington	122 (35.0) 38 (10.9) 130 (37.3) 59 (16.9)	6 (18.8) 10 (31.3) 11 (34.4) 5 (15.6)	9 (30.0) 6 (20.0) 5 (16.7) 10 (33.3)	107 (37.3) 22 (7.7) 114 (39.7) 44 (15.3)	<0.001
Rurality Rural Metropolitan	88 (25.2) 261 (74.8)	12 (37.5) 20 (62.5)	9 (30.0) 21 (70.0)	67 (23.3) 220 (76.7)	0.177
**Nurse attendance**					
Nurse to bed ratio ***Mean (SD)***	1.73 (0.03)	2.02 (0.38)	1.84 (0.22)	1.81 (0.05)	0.577

Notes: *p*-values were derived with ANOVA and Chi-squared tests; MU = Meaningful Use; EHR = Electronic Health Record; SD = standard deviation.

**Table 2 ijerph-19-12525-t002:** Summary statistics of Patient Safety incidence among EHR use.

	Partially Implemented or No EHR *Mean (SD)*	Full-EHR not Receiving MU *Mean (SD)*	EHR that Attests to MU *Mean (SD)*	*p*-Value
**Death Related PSI**				
Death Rate in Low-Mortality Diagnosis Related Groups (DRGs)	1.04 (0.82)	0.10 (0.06)	0.34 (0.04)	0.022
Death Rate among Surgical Inpatients with Serious Treatable Complications	89.21 (15.65)	109.43 (16.42)	124.83 (5.64)	0.222
**Non-Death Related PSI**				
Iatrogenic Pneumothorax Rate (collapsed lung due to medical treatment)	0.28 (0.16)	0.19 (0.05)	8.69 (8.40)	0.897
Postoperative Physiologic and Metabolic Derangement Rate	2.20 (1.84)	0.10 (0.04)	0.49 (0.06)	0.004
Postoperative Respiratory Failure Rate (breathing failure after surgery)	7.54 (4.12)	9.25 (3.48)	8.21 (0.39)	0.810
Perioperative Pulmonary Embolism or Deep Vein Thrombosis Rate (serious blood clots after surgery)	9.21 (4.84)	7.52 (4.03)	4.11 (0.19)	0.007
Postoperative Sepsis Rate	9.44 (3.06)	19.34 (6.97)	8.70 (0.71)	0.004
Postoperative Wound Dehiscence Rate (wounds split open after surgery)	5.90 (4.74)	0.59 (0.24)	1.45 (0.22)	0.006
**PSI-90 Composite Score ***	0.99 (0.03)	0.99 (0.03)	0.95 (0.01)	0.407

Notes: Rates are per 1000 population. Abbreviations: MU, Meaningful Use; EHR, Electronic Health Record; PSI, Patient Safety Indicator. * PSI-90 is a composite score, and not a rate.

**Table 3 ijerph-19-12525-t003:** The impact of EHR use on Patient Safety Indicators.

	Coefficient	Confidence Interval	*p*-Value
**Death Related PSI**			
Death Rate in Low-Mortality DRGs			
Full-EHR not receiving MU	−2.91	−4.31 to −1.51	<0.001
EHR that attests to MU	−0.93	−2.00 to 0.13	0.086
Death Rate among Surgical Inpatients with Serious Treatable Complications			
Full-EHR not receiving MU	0.12	−0.37 to 0.60	0.641
EHR that attests to MU	0.16	−0.22 to 0.53	0.410
**Non-Death Related PSI**			
Iatrogenic Pneumothorax Rate (collapsed lung due to medical treatment)			
Full-EHR not receiving MU	−0.42	−2.29 to 1.44	0.658
EHR that attests to MU	−0.33	−1.72 to 1.07	0.647
Postoperative Physiologic and Metabolic Derangement Rate			
Full-EHR not receiving MU	−2.42	−4.35 to −0.49	0.014
EHR that attests to MU	−1.99	−3.27 to −0.71	0.002
Postoperative Respiratory Failure Rate (breathing failure after surgery)			
Full-EHR not receiving MU	0.68	−0.01 to 1.31	0.053
EHR that attests to MU	0.47	−0.05 to 0.99	0.077
Perioperative Pulmonary Embolism or Deep Vein Thrombosis Rate (serious blood clots after surgery)			
Full-EHR not receiving MU	−0.13	−0.91 to 0.65	0.744
EHR that attests to MU	−0.89	−1.44 to −0.34	0.001
Postoperative Sepsis Rate			
Full-EHR not receiving MU	0.63	−0.31 to 1.56	0.188
EHR that attests to MU	−0.17	−0.86 to 0.52	0.634
Postoperative Wound Dehiscence Rate (wounds split open after surgery)			
Full-EHR not receiving MU	−1.93	−3.43 to −0.43	0.011
EHR that attests to MU	−0.86	−2.02 to 0.31	0.152
**PSI-90 Composite Score**			
**Full-EHR not receiving MU**	−0.02	−0.15 to 0.10	0.701
**EHR that attests to MU**	−0.07	−0.16 to 0.02	0.122

Notes: Reference group: No EHR or Partially implemented HER. Abbreviations: MU, Meaningful Use; EHR, Electronic Health Record; PSI, Patient Safety Indicator. Coefficient is semi-elasticity, where the dependent variable changes by 100*(coefficient) percent for a one unit increase in the independent variable while all other variable in the model are held constant. Model adjusts for minor teaching hospital status, major teaching hospital status, for profit status, state, nurse to staffed bed ratio, and staffed beds.

## Data Availability

The data purchased for this study were provided under Data Use Agreements between the investigators and the data vendors. The data can not be shared, but can be purchased through the data vendors at HCUP-US SID Overview (ahrq.gov) (accessed on 20 January 2016) & American Hospital Association (https://www.ahadata.com/) (accessed on 20 January 2016).

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
