# Peer review of "The Impact of Meaningful Use and Electronic Health Records on Hospital Patient Safety"

_ijerph, 2022, doi:10.3390/ijerph191912525_

Round 1
Reviewer 1 Report
The authors of the current manuscript present the results of a study, aiming to determine the impact of the Electronic Health Records and attesting to Meaningful Use on patient safety indicators. The topic of the manuscript is very important from clinical point of view. The study is well-conducted and the paper is generally well-structured, however I have the following recommendations to the authors for some corrections, that according to me will improve the quality of this article.
- The introduction is too large and detailed: it has to be more succinct. Some parts of the text might be moved to the Discussion.
- There are some parts of the text (in “Analytical approach”) that sound like a discussion, so they have to be moved to “Discussion”
- The size of the “Conclusion” should also be cut (no more that 3-4 short, key sentences).
Author Response
Thank you for your valuable feedback to strengthen our manuscript. We agree and have made the necessary revisions, including:
- We condensed the Introduction to make it more succinct by moving some statements to the discussion and eliminating others. We still provide the references to the statements we eliminated so that the readers can identify the articles for further synthesis.
- We moved statements from the Analytical Approach and inserted them into the relevant places in the Discussion section.
- We have now condensed the conclusion to 4 sentences for conciseness; doing so also reduced redundancies in the discussion.
Reviewer 2 Report
I commend the authors for performing this analysis on EHRs and MUs. The manuscript is overall very well written, presents the introduction, methods, and discussion clearly, and outlines its main objectives well.
I think the main limiting factor is stated briefly in the limitations “Due to the cross-sectional nature to our study, we were not able to establish a causal relationship between EHR use and patient safety”. EHRs are only as good as the information that is inputted into them and how that data drives safety/quality. The burden to put proper terms, DRG keywords, and supporting documentation in the EHR so that a PSI is either activated or not is key, and the quality of the information added to an EHR that drives this may be quite different between institutions (and provider workload, staffing, and other factors could effect this). The second important factor is how ENRs drive quality/safety at these institutions: are there best practice indicators, review committees, or other quality drivers that link EHRs + MUs to actually change/effect patient care? So this all goes back to the link provided by this manuscript being an association, not causation, which limits the overall impact. Study cannot be changed in design to address this concern but perhaps the authors can discuss further in the discussion to drive that point home.
also, I don’t know if it was the screen read out but the first section of table 1 which had the number of hospital beds has to P values that weren’t not lined up with anything, did not know if one was for the overall and then one was for the number of beds but that made me needs to be cleaned up
Author Response
Thank you for your valuable feedback to strengthen our manuscript. We agree and have made the necessary revisions, including:
- Most notably, we recognize that this study has limitations regarding the ability to establish causality from our cross-sectional study design. We do believe this study contributes to preliminary knowledge in the literature that can support future studies that study drivers of patient safety, and ways to leverage EHR systems to do so. We have also included several statements to clarify that EHRs are only as good as the information/functionalities included in the systems. We have included the following statements about the ability of EHRs to improve patient safety in the Discussion: (1) “It is also important to note that providers are reliant on the quality and timeliness of information provided in EHR systems, and the presence of advanced alerts that may indicate patients at risk of adverse events and supporting documentation for care protocols. Research needs to address these nuances in EHR use between healthcare organizations and its impact on individual PSIs.” (2) "EHRs have the potential to improve patient safety, particularly for surgical care by providing timely and meaningful health information that could prevent medical errors or reduce the impact of errors that have been made.”
- We have added a statement in the limitations: “Future studies should account for other drivers of quality that could not be adjusted for in this study, including best practices of healthcare organizations, the presence of review committees, or participation in other programs linked to quality and patient safety.”
- We have fixed the alignment of the P-values in Table 1 for staffed beds.
Reviewer 3 Report
The manuscript describes the impact of Meaningful Use and Electronic Health Records on hospital patient safety.
Patient safety remains a major challenge in the healthcare systems. Recognizing the potential role that health information technology (IT) could play in improving patient safety and quality of care. The results described in this manuscript are consistent with recent studies that find an association between EHR and patient safety indicators using regional or facility-specific data.
Such information is of great interest from hospital leaders perspective, particularly for professionals from those countries in which the health system operates under different economic, organizational and political conditions. However, minor revisions are needed.
For the reader who is not familiar with the regulations in United States health care (hospital policies and procedures), it is quite difficult to follow the presented research concept. Graphical conceptual framework would be helpful (a visual model that assists readers by illustrating how concepts, constructs, themes or processes work).
It is worthwhile to present the definition / description of the following terms: Meaningful Use and Electronic Health Records, Patient Safety Indicators.
It's worth highlighting the fact that especially in the context of surgical care, the EHR can improve patient safety through multiple mechanisms such as providing timely and comprehensive health information that may prevent errors or allow for rapid corrections.
It is necessary to describe in which way hospital patient safety performance was measured (e.g. adverse event indicators).
Table 1 ‘Description of EHR use and Hospital Characteristic’ should be supplemented with other details (e.g. the number of patients admitted to the hospital per year, the number and profile of medical procedures performed, etc.).
Add references to:
· The Health Information Technology for Economic and Clinical Health (HITECH) Act
· Affordable Care Act (ACA)
Author Response
Thank you for your valuable feedback to strengthen our manuscript. We agree and have made the necessary revisions, including:
- I added Figure 1 to the manuscript showing the relationship between EHR use and the regulatory environment as it relates to EHR use, including the HITECH Act and the Meaningful Use program.
- The definition of Meaningful Use is provided on lines 36-42. We added a description on lines 33-34 to state that EHRs are “digitized health records.” The definition of PSI can be found on lines 52-55, “Patient Safety Indicators (PSIs) were developed as metrics for likely preventable patient complications and adverse events following surgeries and procedures, which could provide an opportunity for improvement in healthcare delivery.”
- We have included a statement about the ability of EHRs to improve patient safety: "EHRs have the potential to improve patient safety, particularly for surgical care by providing timely and meaningful health information that could prevent medical errors or reduce the impact of errors that have been made.”
- We provided more clarity in section “2.2. Calculating PSIs”. This methods section now starts with “Patient safety was measured using AHRQ’s PSIs, which are a set of indicators providing information on patient’s potential hospital complications and adverse events following surgeries and procedures.”
- We did not include patient characteristics because we had adjusted for them using the PSI software. The profile of the medical procedures performed would be far too robust with over 5 million patients included in this dataset across three states to fit in the scope of this paper, and not all would be considered hospitalizations that are classified as a PSI. However, Table 2 reports the summary statistics for PSIs for each group. Rates are per 1,000 population. We focused on hospital/organizational level research, and still believe this provides valuable insight into the impact of EHR use and the Meaningful Use program on inpatient quality. We no longer have access to this information due to the DUA agreement.
- We added references for HITECH Act and the ACA.